# Managing well-being in paediatric critical care: a multiperspective qualitative study of nurses' and allied health professionals' experiences

Esra Yeter,[1] Harmeet Bhamra,[1] Isabelle Butcher [iD],[2] Rachael Morrison,[3] Peter Donnelly [iD],[4] Rachel Shaw [iD][5]

[1]College of Health & Life Sciences, Aston University, Birmingham, UK
[2]Department of Psychiatry, Oxford University, Oxford, UK
[3]Paediatric Intensive Care, Birmingham Children's Hospital NHS Foundation Trust, Birmingham, UK
[4]Paediatric Intensive Care Unit, Royal Hospital for Children, Glasgow, UK
[5]Institute of Health & Neurodevelopment, Aston University, Birmingham, UK

**Correspondence to**
Professor Rachel Shaw;
r.l.shaw@aston.ac.uk

## ABSTRACT

**Objectives** It is well evidenced that healthcare professionals working in paediatric critical care experience high levels of burn-out, compassion fatigue and moral distress. This worsened during the COVID-19 pandemic. This work examines the nature of challenges to workplace well-being and explores what well-being means to staff. This evidence will inform the development of staff interventions to improve and maintain staff well-being.

**Design** Qualitative study.

**Setting** Paediatric critical care units in the UK.

**Participants** 30 nurses and allied health professionals took part in online interviews and were asked about well-being and challenges to well-being. Lived experiences of well-being were analysed using interpretative phenomenological analysis.

**Results** Themes generated were as follows: perception of self and identity; relationships and team morale; importance of control and balance and consequences of COVID-19. They focused on the impact of poor well-being on participants' sense of self; the significance of how or whether they feel able to relate well with their team and senior colleagues; the challenges associated with switching off, feeling unable to separate work from home life and the idealised goal of being able to do just that; and lessons learnt from working through the pandemic, in particular associated with redeployment to adult intensive care.

**Conclusions** Our findings align closely with the self-determination theory which stipulates autonomy, belonging and competence are required for well-being. Participants' accounts supported existing literature demonstrating the importance of empowering individuals to become self-aware, to be skilled in self-reflection and to be proactive in managing one's own well-being. Change at the individual and staff group level may be possible with relatively low-intensity intervention, but significant change requires systemic shifts towards the genuine prioritisation of staff well-being as a prerequisite for high-quality patient care.

## INTRODUCTION

Levels of burn-out, compassion fatigue, work-related stress and even post-traumatic stress disorder have reached worrying levels among healthcare professionals generally,[1 2]

### STRENGTHS AND LIMITATIONS OF THIS STUDY

⇒ Nurses and allied health professionals across paediatric critical care (PCC) units in the UK were invited to take part in this research with the support of the Paediatric Critical Care Society.

⇒ A relatively large sample, for a qualitative study, was recruited (n=30), which generated in-depth data describing individuals' lived experiences of well-being and challenges to their workplace well-being.

⇒ Analysis was led by two researchers (EY and HB), with contributions through independent coding and review of themes to reach consensus by the rest of the research team, thus ensuring rigour.

⇒ Findings present an in-depth analysis of individuals' own meaning-making of challenges to their well-being and what workplace well-being looks like to them, which are likely to resonate with other PCC staff.

but levels are higher among those working in paediatric critical care (PCC).[3] Although it has existed for some time, this problem worsened and reached the public consciousness during the COVID-19 pandemic.[4 5] It also caught the attention of the UK government, who concluded it was time for action to reduce burn-out and build resilience among staff working in health and social care.[6] Exposure to patients in distress can negatively impact the well-being of healthcare professionals, as well as those around them.[7] For those working in paediatrics, there are further complexities because of the parents' presence. This adds a unique layer of challenge to the PCC environment.[8] Paediatric nurses develop a relationship with patients' parents, which coupled with extended hospital stays for some patients with complex conditions that have become more chronic as treatment options and survival rates change, can lead to challenges. Challenges might come through strong bonds being developed or to

significant differences of opinion relating to treatment decisions, each of which can lead to moral distress or the blurring of boundaries between personal and professional lives.[8 9]

The threat posed by the psychological distress experienced by PCC staff is not only a problem for their individual well-being but also for the quality of care they are able to provide.[10 11] Compromised well-being has been associated with patient mortality, medical errors, lower patient satisfaction, staff turnover and lower professional effort among staff.[12] In fact, previous cross-sectional studies have found burn-out to be an independent predictor of major medical errors.[13] A recent review which examined levels of burn-out and coping mechanisms employed by staff in paediatric and neonatal intensive care confirmed levels of burn-out were high, but the coping strategies used were not found to be effective.[14] This demonstrates a need for more effective mechanisms to improve well-being and increased self-awareness to enable staff to recognise what works for them. We know that burn-out, stress and psychological distress experienced by PCC staff can lead to them leaving the profession.[15] As such, it is imperative that we understand the nature of challenges to PCC staff's well-being for us to be able to intervene to build resilience from an individual and systemic perspective.

To date, research has focused on measuring the extent of the problem, but Butcher *et al*'s 2023 review[14] and the small amount of qualitative research undertaken to explore workforce well-being in PCC to date[8 16–18] have shown us that further work is required to fully grasp the nature of staff experiences within the context of their lifeworld—their lived experience.[19] To do this, we need to examine in-depth how individuals make sense of what constitutes well-being to them, to understand how they experience challenges to their well-being and how they might be able to integrate well-being strategies into their everyday lives to build resilience at work.

The research question we posed was how do nurses and allied health professionals (AHPs) working in PCC define their own well-being and how do they experience challenges to their well-being?

## METHODS
### Sample
We targeted newly qualified nurses (who worked for less than 5 years as a registered nurse) because existing evidence[18] had suggested that they may struggle more with their well-being than senior colleagues. Equally, we were interested to explore the experiences of senior nurses working as advanced nurse practitioners or well-being leads because they had taken on roles to support others' well-being. Those well-being support roles are sometimes taken on by AHPs and so we extended our invitation to them. There is very little research with AHPs so it was important to us to include them in the sample.

### Procedures
Permission was received from the president to advertise the study through the Paediatric Critical Care Society (PCCS) during January–June 2021. Emails were sent to members and the study was mentioned in regular PCCS meetings. The study was also shared on social media. Interested parties contacted the researchers and informed consent was obtained using Qualtrics (confidential online survey software). Participants were asked to complete a short demographic questionnaire (role, length of time in PCC, age) before being invited to take part in an audiorecorded online semistructured interview (see table 1 for interview schedule). This allowed for the development of an initial rapport with participants, whereby they knew the purpose and role of the researchers in the study. Participants were encouraged to join their interviews in a quiet and safe place on their own. Interviews were transcribed verbatim and any identifiable information was anonymised or removed. Following completion of the interview, participants were sent a debrief form which signposted them to support organisations. A distress protocol was used throughout this study to ensure appropriate safeguarding was in place should any issues of concern for participants or their colleagues be raised. No such issues were raised. Participants were given the option to see their transcripts to ensure they were happy for it to be included.

### Analysis
Interpretative phenomenological analysis (IPA)[20] was chosen to analyse the transcripts. IPA uses an idiographic approach to focus on how individuals make sense of their subjective lived experiences. IPA is informed by phenomenology; its focus is on experience, and it actively involves interpretation from both researcher and participant. This provides a rich source of lived experience themes about well-being in PCC. The researchers continued recruitment until data saturation had been reached.

IPA analysis involves reading transcripts in-depth and undertaking descriptive coding to summarise the key topics raised by participants. Interpretative coding of transcripts is then conducted, which shifts the focus to making sense of what is said about those topics and what matters to participants in relation to how they make sense of their experiences. Interpretative codes are then grouped together to begin to form themes. Initial themes are then clustered together with reference to the research question to form the final set of themes presented in the report.

### Credibility and trustworthiness
The interdisciplinary research team met regularly throughout, with additional reflective meetings of core members during phases of data collection and analysis. Reflexive journals were kept by those leading the analysis. Guidelines for achieving excellence in IPA research were followed.[21] These focus on analytical processes: ensuring there is a compelling narrative with analytic dialogue between data extracts selected; developing a vigorous

**Table 1** Interview schedule

| Topic | Open-ended questions | Prompts |
|---|---|---|
| Introduction | To begin with, can you tell me a bit about yourself? | Tell me about your hobbies/interests. |
| | How would you describe your job? | What does a typical day at work look like for you? |
| | What do you most enjoy about your job? | |
| Well-being | This study focuses on well-being, but that's often quite different for different people. Can you tell me what well-being means to you? | |
| | Does your job in paediatric critical care challenge your well-being in anyway? | Please can you describe those challenges that have affected your well-being? |
| | How does your well-being impact on how you feel about your job? | Can you give me an example? |
| Managing well-being and burn-out | Can you tell me what burn-out means to you? | Has there been a time when you felt burnt out? |
| | | How did you manage this? |
| | How do you balance your personal and professional life? | |
| | Are there specific things you do to manage your well-being? | Can you give me an example? |
| | How do you think your unit manages staff well-being? | |
| | Ideally, how would you like your unit to support your well-being? | |
| Closing | Is there anything else you feel is important that you would like to discuss? | |

experiential account; close analytical reading of data examining participants' meaning-making processes and attending to convergence and divergence to ensure idiosyncrasies are retained alongside aspects of experience that are shared.

### Patient and public involvement
Staff working in PCC were consulted in the design of this study. The study did not involve patients. We did not consult patients or the public.

## RESULTS
### Demographic data
A total of 30 female staff between the ages of 21 and 62 were recruited representing 9 PCC units across England, Wales, Scotland and Northern Ireland, with no participant dropout. Seven AHPs were recruited: 3 clinical psychologists, 1 occupational therapist, 1 dietician and 1 physiotherapist; with 23 nurses, 19 of whom were newly qualified (see table 2). Interviews lasted between 20 and 90 min.

### Interpretative phenomenological analysis
Four main themes were generated (see table 3). Each theme is presented with verbatim transcript extracts. The themes represent constituent elements of well-being developed from participants' interpretation of what well-being means to them. The challenges to well-being shared by participants add depth to our understanding of well-being and the consequences of it not being prioritised among PCC staff.

### Perception of self and identity
This theme explores participants' perception of the self; the idea of 'who you are', one's perceived identity. It demonstrates an inherent connection between the self and how participants make sense of their own well-being. We examine how the struggle to maintain well-being and an integrated sense of self can be challenged by a busy work life.

**Table 2** Participant characteristics

| Role | N | Length of time in PCC | n | Age (years) | n |
|---|---|---|---|---|---|
| Band 5 | 20 | Up to 1 year | 9 | 20–25 | 12 |
| Band 6 | 3 | 1–3 years | 11 | 26–30 | 9 |
| Band 7 | 2 | 3–5 years | 2 | 31–35 | 0 |
| Band 8 | 0 | 5–10 years | 4 | 36–40 | 3 |
| Clinical psychologist | 1 | 10+ years | 4 | 41–45 | 3 |
| Occupational therapist | 1 | | | 46–50 | 2 |
| Physiotherapist | 1 | | | | |
| Total | 30 | | | | |

PCC, paediatric critical care.

**Table 3** Analytical themes

| Theme | Description |
|---|---|
| Perception of self and identity | The relationship between perceived self and well-being. How participants reconnect with the self that is lost within the busy work–life. |
| Staff relationships and team morale | How staff relationships shape a sense of group identity and performance in PCC. Distorted perceptions of staff views and its impact on the perceived value of self. |
| Importance of control and balance | How the concepts of control and balance become meaningful in the pursuit of well-being. |
| Consequences of COVID-19 | The impact of staff re-deployment on well-being. Adult ICUs versus PCCs. |

ICUs, intensive care units; PCC, paediatric critical care.

Feelings of guilt emerged through a perceived expectation to go above and beyond in one's work, a feeling which contributed to how they constructed their own sense of self. Self is a psychological concept, which refers to one's sense of identity, uniqueness and self-direction, in comparison to others in our social world.[22] The social systems we live and work in, together with the people that populate them, can impact strongly on our constructions of our sense of self. Among the newly qualified nurses in this study, guilt was a common feeling which made them feel differently about themselves. They felt guilty when considering taking time off work to take care of themselves and their well-being. Protecting the self, or engaging in self-care, is important to retain a strong sense of self. Although participants acknowledged the importance of self-care, they carried guilt associated with time spent away from work and patients who needed them. The extract below exemplifies how the sense of self is lost within through these expectations to an extent that time away from work leaves them feeling 'guilty':

What's going on inside and you almost… you do this job because you're caring and you're you want to help and then feel guilty because actually you need to take that time out… for yourself because actually you know you're gunna [/going to/] head for… having more time off work if you don't take that time (Participant 6, newly qualified nurse)

Working in PCC one is 'expected' to immediately recover from the challenges faced at work. Participants held the belief that there is an expectation 'to just pull yourself together' and to automatically know how to manage challenging events on the unit:

I think that… they expect you to just pull yourself together and get on with it. (Partcipant 1, newly quali-fifed nurse)

Nature and water were commonly reported as a way of reconnecting with the self that is lost within the busy work

life. Participants found well-being when they were close to nature. They highlighted the importance of being with the ordinary features of the world to find oneself. Well-being appears to be a moment of slowing down and simply existing with oneself. The feeling of being still in the world outside contrasts with the pressures and responsibilities on the unit, which points to the value of this calmness for their well-being:

it just feels kind of… freeing and like just looking at the water just falls freely into the water the noise it made and it was just really peaceful then and… I dunno I just sat like I sat there for really for about 10 minutes cuz you could go behind the waterfall and I probably sat there for about 10 minutes and it just felt like… very… very at peace with myself and yeah it was just a lovely… a lovely day… (Participant 30, Band 6 nurse)

Being away from 'other people' was important for participants' well-being. Working in PCC involves contact and communication with patients and staff, which can be overwhelming. Nature allows for the separation from other people and to be present in the moment:

so I think there's something about erm… there are no people [laughs] in it [both laugh] erm [I: yeah] the thought of being there and kind of on your own erm… beautiful setting I really like being kind of in trees you know around trees and erm… just kind of reminded me I guess of my morning run where it looks really beautiful and you're kind of on your own and it just feels really… really good er and I think yeah I… the ones where there are other people in I was just like that to me isn't wellbeing I think I dunno if it's because my job is so with other people [I: yeah] and at home I'm with other people all the time I just really value… erm… for my wellbeing I know I need that time on my own to kind of recharge… (Participant 21, Clinical psychologist)

However, for some participants when asked what well-being meant to them, they focused on prioritising the well-being of others indicating a reticence or unfamiliarity with putting themselves first:

wellbeing I think looking after our elderly… is er… is important unfortunately I do not have any of my grandparents… but I have my mum she's not that old… but if I look at this picture then… she's one of the things I think about and erm… and I also think about er- so I used to work in a nursing home as a nurse looking after elderly and… it had its own beauty and difficulties as well but… yeah I think erm… looking after the generation before us… is erm… is a part of wellbeing [I: yeah] I think at least for me… (Participant 24, Band 5 nurse)

When not appropriately supported, the stress of working in PCC can impact participants' constructions of the self. Feelings of guilt derived from a perceived expectation to

just be able to get on with things without proper psychological care can erode the sense of self. The grounding and healing effect of nature allows participants to reconnect with that lost sense of self. Well-being is about going back to the ordinary and being free of distressing responsibilities—just simply existing with what is healthy for the soul, mind and body.

### Staff relationships and team morale

Many participants spoke of their colleagues as friends and felt comforted by others who 'felt the same' as them. Feeling connected to colleagues helped newly qualified nurses in particular feel less isolated. They also described a sense of security and trust with the colleagues they had befriended. This was especially significant when they had a particularly stressful day or when patients died:

> And that I'd, I genuinely consider friends… and I think if I didn't have that, if I was just working with people that I was like friendly with but I didn't really like them or. I don't know I couldn't like, I feel like I couldn't have like a laugh with I think I probably wouldn't like work as much as I do. (Participant 2, Newly qualified nurse)

Having friends in PCC to rely on for moral support enabled nurses to cope. However, not all staff felt this support, in fact, some reported that management on some units were not as supportive of their well-being as their peers. Staff felt they were unable to be vulnerable and express their feelings to their senior colleagues. This lack of perceived approachability of senior colleagues created a hierarchical divide. This resonates with the perceived 'normalised culture' of struggling in silence:

> I don't think they handle it good at all, I don't think that they take it seriously, Erm… I don't think that… they… really… care as a management unit side. (Participant 1, Newly qualified nurse)

Staff relationships can impact on participants' socialised sense of self, and when those relationships are not supportive, it can lead to individuals questioning their self-worth and perceived value on the unit. Self-doubt was a commonly reported feeling among PCC staff, something influenced by how they perceived themselves through the eyes of other staff, as exemplified in the extract below:

> I'm the alone voice shouting and and you kinda think am I making fuss about something that… it's unnecessary that nobody else really cares about nobody else thinks it's important erm and there are… there are moments when I kind of go erm you know am I actually making a difference? erm… yeah that you know and sometimes you think actually I'm not (Participant 22, Band 7 nurse)

The perceived expectation among colleagues to be working around the clock also contributed to self-doubt and an eroding sense of self. Some AHPs felt separated from their nursing colleagues because they do not work

typical shift patterns. Instead, they usually work standard office hours (Monday to Friday, 09.00–17.00). This different working structure contributed to irrational thoughts about the value of their role:

> … I think it's hard when you work on ev- you know like in PCC because [I: yeah] the other staff are there all the time I mean they're not there all the time but there are people there all the time it's kinda kind 24/7 so there's always a slight sense of people see me as not doing enough cuz I'm just kind of… certain hours I don't do weekends and I'm not there on bank holidays or or whatever and erm… I think er I've yeah I don't I don't I don't know that actually people think that [laughs] that's perhaps my perception but it always feels as though you have to work hard to kind of justify your existence. (Participant 21, Clinical psychologist)

In summary, staff relationships within PCC can significantly impact participants' well-being, which can be experienced as self-doubt and a distorted sense of self. Whether or not there is evidence of others undervaluing the roles of AHPs, for example, the feeling is experienced as real. As such, self-doubt and poor self-worth can act as a threat to staff well-being.

### Importance of control and balance

Perceived sense of control and achieving balance is instrumental in achieving workplace well-being. There was variation in participants' ability to achieve a sense of control in PCC. Good perceived control within their role boosted their well-being.

Some newly qualified nurses found it difficult to strike a balance between work and home life. If well-being is conceived in economic terms, newly qualified nurses spent all their emotional resources to maintain well-being in the face of adversity while at work. This meant there was nothing left in the pot for their home life—they had 'nothing left to give' outside of work, which could lead to compassion fatigue:

> Um compassion fatigue to me means that… at work um we spend so much compassion we… I mean we immerse ourselves in the lives of our patients….And their families and their experiences… and then you go home and… you basically have well no-nothing left to give because it's all spent at work. (Participant 7, New qualified nurse)

Others discussed their own strategies for achieving a work-home balance. One method of achieving balance was through reflective practice. Being reflective can bring closure to a work shift; a way to switch off and transition to their personal life. This does take practice and conscious effort. As we see in the extract below, not engaging in reflection could mean work-related distress can penetrate one's dreams and lead to poor sleep quality, which can negatively impact on well-being:

And not really looking at yourself and then when you come home if you don't use that time to reflect and think about….Which I sometimes don't then I tend to dream about it. (Participant 9, Newly qualified nurse)

Another successful strategy for one participant was their physical exercise routine and weight loss programme. She felt she was able to restore her sense of lost confidence experienced when working in PCC sometimes by exacting control over her physical health, which helped to build her mental health. This was especially true for this individual because it was during the uncertainty and fast-moving policy changes during the COVID-19 pandemic:

…with this weight loss and the exercises and erm… I think the reason why it became so important to me… doing these things on a regular basis because… that was the only thing I felt I had control over… every aspect of life related to work I lost we lost all of the controls we didn't know… where we gonna be what our days gonna be like it wasn't like a normal day on PCC or anywhere else… we could not do anything about it… as you said changes were so frequent… new guidelines… new new things to follow new equipment new rules everything everyday was a challenge… but at home with my exercises I could control those things… (Participant 24, Band 5 nurse)

The importance of balance in maintaining well-being was further emphasised because of the emotional exhaustion experienced working in a busy PCC environment. Being able to compartmentalise this work-related emotional distress from one's home life enabled staff to take control and achieve balance to help maintain a healthy mental well-being:

I think a lot of wellbeing to me is more around just that balance and having that balance between… cuz obviously… the PCC is such a stressful environment you come across a lot of emotional things… erm… and it's often hard to not let that affect you once you've gone home so it's just trying to have that balance between… being at work and…when you're at home (Participant 28, Occupational therapist)

Control and balance were identified by all participants as important for their well-being. Those who felt they had control over their work and managed to maintain a balanced routine appeared to be less distressed than others.

## Consequences of COVID-19

The data for this study were generated during the COVID-19 pandemic so inevitably participants discussed how their experience of the pandemic impacted on their well-being, which does offer relevant learning for future management of well-being. Many participants described being more aware of needing to pay attention to their well-being during the pandemic:

I could feel myself kind of… physically and mentally exhausted um so I started using meditation because I wasn't sleeping very well… Um and I've put that down to Covid and the pressures around work the pressures around home life… and the pressures around just general society in general really (Participant 6, Newly qualified nurse)

The pandemic was experienced as a challenge to well-being. For this newly qualified nurse, learning how to deal with the pressures of the working environment was heightened because of COVID-19. Nevertheless, it meant that the introduction of strategies to improve well-being, in this case practising meditation to help with sleep, came early in the career path, thus embedding the need to prioritise well-being.

One of the most challenging consequences of the pandemic for PCC staff was 'staff redeployment' to adult intensive care units (ICUs) and in some cases the admission of adult patients to PCC units. This happened largely during the first wave of the pandemic, due to limited bed capacity in adult ICU (Between March and June 2020 in the UK, a national lockdown was introduced during which people were only permitted to leave their household for very limited purposes: (1) Shopping for basic necessities, for example, food and medicine, which must be as infrequent as possible. (2) One form of exercise a day, for example, a run, walk or cycle—alone or with members of your household. (3) Any medical need, including to donate blood, avoid injury or illness, escape risk of harm or to provide care or to help a vulnerable person. (4) Travelling for work purposes, but only where you cannot work from home. UK government guidance during lockdown is available here: https://www.gov.uk/government/publications/full-guidance-on-staying-at-home-and-away-from-others/full-guidance-on-staying-at-home-and-away-from-others#stopping-public-gatherings). This change in work environment, colleagues and differences in practices, techniques and equipment from child to adult care was experienced as challenging, especially for those who were not trained to work with adults. One participant described being 'thrown into' adult care, which was challenging both emotionally and intellectually, and in turn impacted on well-being.

we were thrown into [inaudible] we had adults instead of… paediatrics… erm so it was it was… a really ?challenging? time for a lot of people everyone was out of their comfort zone it wasn't something that any of us have been trained to do even and we were kind of learning as we were going along which… to any clinician… is… doesn't feel right does it so it's erm… [inaudible] erm so that out of control… compromises your wellbeing because… you… literally have to concentrate on trying to do the best job and the best ?thing for? These… horrendous situations for patients so… (Participant 23, Band 7 nurse)

Witnessing other staff struggle with these challenges coupled with their continued redeployment was morally distressing for staff not included in these measures:

> we weren't included in the count to move… and I found that really really hard to… try–just to see a colleague's crying and struggle and then still have to say to them "actually you're needed elsewhere" erm… and that was really difficult because… you know I didn't know what to say to them really because… I really sympathised with… the position they were in but equally like it was their turn… (Participant 30, Band 6 nurse)

The pandemic also led to increased staff absence due to sickness and self-isolation. This meant there was an increased need to cover sickness as well as covering for increased admissions of adult patients. This created pressure above and beyond what PCC staff face typically, with reports of some staff quitting their jobs as a result, which further exacerbated the distress experienced on the units:

> It's just adding to that… and I think a lot of us are getting really upset and even I think some people have genuinely left uh because we're having to help out so much. (Participant 23, Band 7 nurse)

Being redeployed to adult ICU caused some participants a lot of distress. Staff who were redeployed experienced distress due to perceived lack of competence in adult care, but also the significant unfamiliarity with the team, environment, tasks and equipment. Staff who were not redeployed witnessed these difficulties and began to internalise the distress of the redeployed staff, leaving them feeling guilty for escaping their fate, leading to moral distress. These findings further highlight how changes for some can impact well-being across the whole unit.

## DISCUSSION

This study explored well-being experiences with a sample of nurses with varying levels of experience and AHPs working in PCC. Some newly qualified nurses in our sample described PCC as a 'toxic environment' where they felt compelled to 'just get on with it' instead of raising their feelings of vulnerability with senior colleagues. This created a felt hierarchical divide between newly qualified and more senior nursing staff, which further exacerbated the unrealistically high expectations newly qualified nurses placed on themselves.

Others reported experiences of sleepless nights and anxieties surrounding work, which are common symptoms of compassion fatigue[23]; symptoms that have only got worse during the COVID-19 pandemic.[24] Participants also described that the 'cost of caring'[25] meant there was little compassion left for themselves or their family outside of work.

It was clear from participants' accounts that they had not received training or mentorship to guide them in self-care or in developing constructive coping mechanisms for managing challenges to workplace well-being. Providing education in these areas has been shown to reduce the risk of burn-out and compassion fatigue[26] and thus could contribute to improved staff well-being.

Self-reflection is required to develop self-awareness and for individuals to identify what would work to improve their well-being. It is also essential to professional development. As such, self-reflection has been identified as an essential skill for PCC staff which can also reduce risk of burn-out,[26] but it does require investment in training and/or mentorship.[27] Building these skills in mandatory training prior to qualification and maintaining them through continued professional development may help to reduce the number of those who leave the profession or switch specialty away from PCC due to burn-out.[28]

Nurses' and AHPs' discussions around what well-being means to them and how they manage it revealed that grounding oneself in nature and being immersed in the ordinary features of the natural world, for example, by practising mindfulness, were central. There is now growing evidence that being in nature has a positive impact on our wellbeing. This feeling of being 'at home' with oneself and the world sits at the heart of what it means to be human; it offers us a feeling of peace but opens up a host of potentialities which might be invisible when we're feeling 'stuck' or when our sense of self is challenged.[29–31]

Redeployment to adult intensive care led to moral distress, particularly for senior staff making decisions to move their staff to this unfamiliar environment, supporting existing evidence.[32] It is understandable that it was especially difficult to manage staff well-being during the COVID-19 crisis, however, the transferable lesson here is that limitations in competence lead to depleted confidence, which in turn increases the risk of moral distress and poor well-being. To improve and maintain healthy well-being, staff need to feel in control of their work, feel accepted as part of a team and have appropriate competence levels.[33]

Finding control and maintaining a work–life balance contributed to staff well-being. Those who had a blurred home–work balance reported feelings of distress while those who had more control and a good balance reported experiencing better well-being. Achieving work–life balance is a strategy to reduce stress. Stress is a common consequence of feelings of powerlessness,[34] like those experienced when the ability to control one's work–life balance is lost. A central message throughout this analysis is that PCC staff need to develop strategies for managing their stress, they need support when they become distressed, and they need to be supported in proactively managing their own well-being. Of course, for this to be achieved, the system in which they work needs to prioritise staff well-being, recognise it as a fundamental prerequisite for maintaining a healthy workforce and for good quality patient care.

## Limitations

All those who participated identified as female. Further research including male PCC staff members would, therefore, be beneficial. This study was based in the UK; research with PCC staff in other countries would benefit the evidence base. Furthermore, this work was conducted during the COVID-19 pandemic which could mean some aspects of the findings are specific to this time period. However, the prevalence of burn-out and compassion fatigue among PCC staff pre-existed the pandemic[3] and many of the challenges to well-being highlighted by staff are commensurate with the psychological theory of self-determination.[35] As illustrated above, this helps us identify the requirements for staff to experience workplace well-being. This alignment with theory strongly suggests the longevity of findings. Further research postpandemic will help confirm this.

## Implications for clinical practice

Enhancing staff well-being requires systemic change to create an environment which prioritises well-being—these kinds of change are slow. While pursuing those systemic changes we can make smaller, more local changes designed to improve the everyday lived experience of PCC staff. As we have seen, providing staff with self-reflection skills can reduce risks of burn-out. Providing supportive structures—through social support groupings or peer support programmes—can help create a psychologically safe environment and foster a team identity to help staff feel comfortable in discussing their vulnerabilities with senior colleagues.[36] A buddy or mentoring system for newly qualified nurses would be beneficial. We need to encourage staff to identify what well-being means to them so they can create their own 'tailor-made' well-being plans which fit with their lifestyle and they enjoy. Introducing conversations about well-being into the work environment with images of nature[16] might help give staff a boost and remind them that managing their well-being is crucial to successful working lives. Furthermore, the formal introduction of well-being conversations within professional development reviews would help prioritise well-being within career development. Finally, it is important to recognise and express gratitude for staff's efforts, especially when they go the extra mile to support colleagues or go out of their way to ensure families have a better experience.

## Implications for future research

We need to gather rigorous evidence to evaluate the impact of well-being interventions to determine whether they can show positive changes in staff well-being. The longer-term project is to demonstrate the cost-effectiveness of staff well-being interventions and their impact on outcomes including staff retention, sickness absence, and even patient safety measures and patient outcomes.

## CONCLUSION

There are clear indications that PCC staff need support to improve and maintain their well-being. Pressures leading to burn-out, compassion fatigue and moral distress have worsened with the COVID-19 pandemic. Now is the time to act to care for the carers. Staff well-being interventions can be simple, but their focus needs to be on reflective communication, making sense of challenging and distressing experiences, fostering a psychologically safe environment where staff feel free to express vulnerabilities and to benefit from their shared wisdom. Prioritising staff mental and physical health is the only solution that will make a genuine difference to workplace well-being.

**Acknowledgements** We are grateful to all who gave their time to take part in this research and to James Fraser, Past President of Paediatric Critical Care Society and the Paediatric Critical Care Society's Wellbeing Group for supporting recruitment.

**Contributors** Guarantor of the work is RS (health psychologist). The research idea was conceptualised by RS, RM (advanced nurse practitioner), PD (medical consultant) and IB (psychology researcher,). EY (health psychology) and HB (health psychology) designed the study, recruited participants, collected and analysed all the data. All researchers, besides PD, are female. Supervision and appropriate training were provided by RS with support from IB. EY, HB and IB led the writing of the manuscript. All authors contributed to and approved the final version.

**Funding** The study received no direct funding, but it was sponsored by Aston University and supported by the Paediatric Critical Care Society. Time for Isabelle Butcher was funded by Birmingham Women's and Children's Hospital Charity Paediatric Intensive Care funds, Ref: 37-6-124.

**Competing interests** None declared.

**Patient and public involvement** Patients and/or the public were involved in the design, or conduct, or reporting, or dissemination plans of this research. Refer to the Methods section for further details.

**Patient consent for publication** Consent obtained directly from patient(s).

**Ethics approval** All methods were carried out in accordance with the Declaration of Helsinki and were approved by Aston University Research Ethics Committee. All participants gave their informed consent to take part in the research.

**Provenance and peer review** Not commissioned; externally peer reviewed.

**Data availability statement** No data are available. The datasets generated in the current study are not publicly available due to the challenges in the successful anonymisation of healthcare professionals from small groups within small hospital departments.

**ORCID iDs**
Isabelle Butcher http://orcid.org/0000-0003-2915-8269
Peter Donnelly http://orcid.org/0000-0003-1133-157X
Rachel Shaw http://orcid.org/0000-0002-0438-7666

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
