## [Reviewer comments · BMJ Open]

ARTICLE DETAILS

TITLE (PROVISIONAL)	Managing wellbeing in paediatric critical care: a multi-perspective qualitative study of nurses' and allied health professionals' experiences
AUTHORS	Yeter, Esra; Bhamra, Harmeet; Butcher, Isabelle; Morrison, Rachael; Donnelly, Peter; Shaw, Rachel

VERSION 1 – REVIEW

REVIEWER	Menzies, Julie Bristol Royal Hospital for Children, United Hospitals Bristol and Weston NHS Foundation Trust, Paediatric Intensive Care Unit
REVIEW RETURNED	16-Mar-2024

GENERAL COMMENTS	Thank you for the invitation to review this paper by the SWell study team. The team have undertaken a large body of work within Paediatric Critical Care and this paper adds further evidence the challenges staff face working within this area. Wellbeing is very topical, but where this paper makes a unique contribution is that participants defined what workplace wellbeing meant to them and strategies that support this. The authors acknowledge the context of the study which was conducted during the pandemic and other appropriate limitations. Most of comments reflect minor tweaks or additional reporting which I think will enhance the paper, particularly the processes involved with using IPA. I look forward to seeing this paper in print. Introduction:  • 'This adds a unique layer of challenge in the PCC environment[8]Paediatric nurses develop a relationship with patients' parents, which, coupled with extended hospital stays for some patients with complex conditions that have become more chronic as treatment options and survival rates change, can lead to strong bonds being developed which in turn can lead to moral distress or the blurring of boundaries between personal and professional lives.[9]' Can you break up this sentence – very long currently? Also what about the other side of this- when tensions form and the challenges of caring for families when there is a strained relationship / lack of trust?  • 'Evidence shows that the quality of distress experienced by PCC staff is distinctive from that felt by healthcare staff working in adult care because of the vulnerability of the population they work with.[14]' I'm not sure this sentence fits well here or is needed?  • 'A recent review which examined levels of burnout and coping mechanisms employed by staff in paediatric and neonatal intensive care confirmed levels of burnout were high, but staff were not using
--

	effective coping strategies and so were unable to improve their wellbeing. [15]' The phrasing of this doesn't sit quite right for me- the way it's phrased it sounds as if staff are to 'blame' for not improving their own well being. I haven't read the paper but is there more that could be shared about why they weren't using effective coping strategies? Or do you need to expand on why. Is it sufficient to just highlight that there are high rates of burnout?  • Participants were given the option to see their transcripts and edit if they wished to. Unsure about the phrasing of edit as they wished? Also I don't think this specifically enhances the trustworthiness of results which comes more from reviewing the researchers interpretation of the results? Method  • advertise the study through the Paediatric Critical Care Society (PCCS) during January to June 2021. Not quite clear how the society advertised eg was this email to all members or promotion on their website etc  • Newly qualified nurses (worked for less than five years as a registered nurse), senior nurses working as Advanced Nurse Practitioners or wellbeing leads and allied health professionals The justification for these inclusion criteria is not quite clear. The authors do specifically state little is known about AHPs but there is no justification for the targeted focus on Newly qualified and ANPs or wellbeing leads. Is there something specific to these groups that made them the focus? What about nurses 5+ years?  • I feel a little more detail about the step by step processes within IPA analysis would be useful. I appreciate there is a word limit but it feels very light on detail about the steps the researchers took and how they arrived at 4 core themes. • Reviewing elements which increase the credibility and trustworthiness of the study I can see the explicit statement: 'Analysis was led by two researchers (EY and HB), with contributions through independent coding and review of themes to reach consensus by the rest of the research team, thus ensuring rigour.' And 'Participants were given the option to see their transcripts and edit if they wished to.' I think the reporting would be enhanced by slightly more detail about how credibility is demonstrated in IPA and more detail on how this was ensured within the study. Results  • A total of 30 female staff between the ages of 21-62 were recruited from across a range of UK PCC units I wondered if it might be useful to report how many units were represented? Important perhaps to demonstrate that this is not related to issues / challenges within one unit or one geographical area?  • Table 1: Participant characteristics I'm unsure about whether you need to report on age. I mention this because I can identify one participant by their job role and age so I'm a little concerned about the anonymity. I don't think you consider age in the analysis and interpretation of results- only the level of experience, so I feel this could be omitted.
--	--

REVIEWER	Slater, Penny Children's Health Queensland Hospital and Health Service
-----------------	---

REVIEW RETURNED	19-Mar-2024
-------------

GENERAL COMMENTS	This is an interesting paper on paediatric critical care staff perceptions of challenges to and maintenance of, their wellbeing. Introduction - There seems to be a mismatch of the research question that was posed and the themes and sub themes raised in the results. Method - Not having the interview questions that were asked in the study does not allow an understanding of the focus of the research question. These questions need to be included. Analysis – more information is needed on how themes were constructed eg independent coding, review of themes and reaching consensus within the methods section. Results - It would be useful to quantify the participants who aligned to a particular theme or sub-theme so the reader understands how prolific that theme was in the sample. This could be done in a more expansive version of Table 2 that includes the sub themes. The linking of the perception of self and identity with guilt for taking time off work for self care needs some more explanation and how that was demonstrated in the interviews. Similarly, the linking of nature and water with finding oneself. Discussion - Self reflection is mentioned as an essential skill, but it is only loosely linked with the results. You probably need to make it clearer if you are linking that to how staff found sitting in nature helpful. However, the only quote we have about this seems to be about being mindful rather than actively self reflecting. Typos and tense corrections: Page 3 line 39 – change feel to felt. page 12 line 52 “slight”
--

VERSION 1 – AUTHOR RESPONSE

Reviewer 1’s comments	Authors’ response
Introduction 'This adds a unique layer of challenge in the PCC environment[8]Paediatric nurses develop a relationship with patients’ parents, which, coupled with extended hospital stays for some patients with complex conditions that have become more chronic as treatment options and survival rates change, can lead to strong bonds being developed which in turn can lead to moral distress or the blurring of boundaries between personal and professional lives.[9]' Can you break up this sentence – very long currently? Also what about the other side of this- when tensions form and the challenges of caring for families when there is a strained relationship / lack	Thank you for this. We have broken up the sentence and included reference to the multiple ways relationships with parents might impact staff, e.g., in differences in treatment decisions.

of trust?	
'Evidence shows that the quality of distress experienced by PCC staff is distinctive from that felt by healthcare staff working in adult care because of the vulnerability of the population they work with.[14] I'm not sure this sentence fits well here or is needed?	We have deleted this (& the associated reference as it is no longer cited).
'A recent review which examined levels of burnout and coping mechanisms employed by staff in paediatric and neonatal intensive care confirmed levels of burnout were high, but staff were not using effective coping strategies and so were unable to improve their wellbeing. [15] The phrasing of this doesn't sit quite right for me- the way it's phrased it sounds as if staff are to 'blame' for not improving their own well being. I haven't read the paper but is there more that could be shared about why they weren't using effective coping strategies? Or do you need to expand on why. Is it sufficient to just highlight that there are high rates of burnout?	We have changed the wording of this. It is more that the measures aren't very good at detecting whether coping strategies are effective. So we have changed the wording to take that potential interpretation of fault as that is not what we intended.
Participants were given the option to see their transcripts and edit if they wished to. Unsure about the phrasing of edit as they wished? Also I don't think this specifically enhances the trustworthiness of results which comes more from reviewing the researchers interpretation of the results?	We have changed this as we agree with you. It's more that participants were given a chance to review their transcription & its inclusion in the study, given the sensitivity of material.
Methods	
advertise the study through the Paediatric Critical Care Society (PCCS) during January to June 2021. Not quite clear how the society advertised eg was this email to all members or promotion on their website etc	We have added in detail here.
Newly qualified nurses (worked for less than five years as a registered nurse), senior nurses working as Advanced Nurse Practitioners or wellbeing leads and allied health professionals The justification for these inclusion criteria is not quite clear. The authors do specifically state little is known about AHPs but there is no justification for the targeted focus on Newly qualified and ANPs or wellbeing leads. Is there something specific to these groups that made them the focus? What about nurses 5+ years?	We have added our rationale here – we wanted to get perspectives from both ends of the spectrum – those with less experience and those who have much more experience & have clearly made a commitment to supporting their colleagues' wellbeing. Thanks for helping us improve this section.
I feel a little more detail about the step by step processes within IPA analysis would be useful. I appreciate there is a word limit but it feels very light on detail about the steps the researchers took and how they arrived at 4 core themes.	We've included the main steps in the process of IPA – thanks for suggesting this as we realise BMJ Open's readership may not be as familiar with this as with other qualitative methods.
Reviewing elements which increase the credibility and trustworthiness of the study I can see the explicit statement: 'Analysis was led by two researchers (EY and HB), with contributions through independent coding and review of themes to reach consensus by the rest of the research team, thus ensuring rigour.' And 'Participants were given the option to see their	We have added a section to the Methods describing creditability & trustworthiness, as we agree it is important to emphasize these aspects of the study. This helps address these points.

transcripts and edit if they wished to.’ I think the reporting would be enhanced by slightly more detail about how credibility is demonstrated in IPA and more detail on how this was ensured within the study.	
Results A total of 30 female staff between the ages of 21-62 were recruited from across a range of UK PCC units I wondered if it might be useful to report how many units were represented? Important perhaps to demonstrate that this is not related to issues / challenges within one unit or one geographical area?	We recruited from 9 PCC units with representation from England, Wales, Scotland & Northern Ireland so have included those details now.
Table 1: Participant characteristics I’m unsure about whether you need to report on age. I mention this because I can identify one participant by their job role and age so I’m a little concerned about the anonymity. I don’t think you consider age in the analysis and interpretation of results- only the level of experience, so I feel this could be omitted.	We have altered this table completely, thanks for suggesting this. This is now Table 2.
Reviewer 2 Introduction - There seems to be a mismatch of the research question that was posed and the themes and sub themes raised in the results.	We have made the link between the research question & themes by adding in some explanation about the status of the themes. The research question is about understanding how staff make sense of their own wellbeing & challenges to their wellbeing. The themes indicate elements that are required for participants to experience wellbeing. Their accounts about what challenges wellbeing helps us then to understand the implications of poor wellbeing or wellbeing being neglected. We hope this helps make the link explicit.
Method - Not having the interview questions that were asked in the study does not allow an understanding of the focus of the research question. These questions need to be included.	We have added the interview schedule to a new table (table 2). Thanks so much for this suggestion as it brings it in line with what has come to be expected in a qualitative paper.
Analysis – more information is needed on how themes were constructed eg independent coding, review of themes and reaching consensus within the methods section.	Reviewer 1 also requested this – see response above.
Results - It would be useful to quantify the participants who aligned to a particular theme or sub-theme so the reader understands how prolific that theme was in the sample. This could be done in a more expansive version of Table 2 that includes the sub themes.	We haven’t done this because it doesn’t align with IPA as a methodology. We have added in quality criteria for IPA which help situate what makes a good quality IPA study. Readers will be able to see from that that frequencies of people contributing to a theme doesn’t feature in IPA analysis. Having said that, we haven’t presented themes unless there is representation from across the participant sample. It is possible sometimes to include themes in IPA which only 1 or 2 participants contribute to, but that isn’t the case here.
The linking of the perception of self and identity with guilt for taking time off work for self care needs some more explanation and how that was demonstrated in the interviews. Similarly, the linking	We have explained these points more fully in the discussion. Thanks for highlighting that further explication was required. We hope we have addressed this issue.

of nature and water with finding oneself.	
Discussion - Self reflection is mentioned as an essential skill, but it is only loosely linked with the results. You probably need to make it clearer if you are linking that to how staff found sitting in nature helpful. However, the only quote we have about this seems to be about being mindful rather than actively self reflecting.	We have made a more explicit link between self reflection and our findings about wellbeing where this is raised in the discussion. Hopefully that makes things more coherent.
Typos and tense corrections: Page 3 line 39 – change feel to felt. page 12 line 52 “sli:ght”	Thanks for these. We didn’t find a ‘feel’ that should have been ‘felt’. The use of : indicates length of time spent saying the word, but we’ve deleted it since we don’t really use any other transcription symbols like this.

VERSION 2 – REVIEW

REVIEWER	Menzies, Julie Bristol Royal Hospital for Children, United Hospitals Bristol and Weston NHS Foundation Trust, Paediatric Intensive Care Unit
REVIEW RETURNED	03-May-2024

GENERAL COMMENTS	Thank you for the opportunity to re-review this paper and to the authors for their response. It is clear from the summarised list that the authors have responded in full to comments from both reviewers and the editorial team. Thank you for addressing all comments so thoroughly, particularly around adding clarity about the method, inclusion criteria and the addition of the interview schedule. The paper is much improved and I have no further comments to make. I look forward to seeing the paper in print.
---

REVIEWER	Slater, Penny Children's Health Queensland Hospital and Health Service
REVIEW RETURNED	10-May-2024

GENERAL COMMENTS	The revised paper has addressed the factors brought up in the first round of review with the exception of providing some quantities in Table 3 of how many respondents provided feedback related to each of the themes. I feel that these figures would add credibility to the development of themes and their discussion in the paper.
---